# A Novel Temporal Network-Embedding Algorithm for Link Prediction in Dynamic Networks

**DOI:** 10.3390/e25020257

**Published:** 2023-01-31

**Authors:** Khushnood Abbas, Alireza Abbasi, Shi Dong, Ling Niu, Liyong Chen, Bolun Chen

**Affiliations:** 1School of Computer Science and Technology, Zhoukou Normal University, Henan 466000, China; 2School of Engineering and IT, The University of New South Wales (UNSW), P.O. Box 7916, Canberra, ACT 2610, Australia; 3School of Software Engineering, Zhoukou Normal University, Zhoukou 466000, China; 4School of Computer Science and Engineering, Huaiyin Institute of Technology, Huaian 223003, China

**Keywords:** graph representation learning, node embedding, temporal link prediction, temporal networks

## Abstract

Understanding the evolutionary patterns of real-world complex systems such as human interactions, biological interactions, transport networks, and computer networks is important for our daily lives. Predicting future links among the nodes in these dynamic networks has many practical implications. This research aims to enhance our understanding of the evolution of networks by formulating and solving the link-prediction problem for temporal networks using graph representation learning as an advanced machine learning approach. Learning useful representations of nodes in these networks provides greater predictive power with less computational complexity and facilitates the use of machine learning methods. Considering that existing models fail to consider the temporal dimensions of the networks, this research proposes a novel temporal network-embedding algorithm for graph representation learning. This algorithm generates low-dimensional features from large, high-dimensional networks to predict temporal patterns in dynamic networks. The proposed algorithm includes a new dynamic node-embedding algorithm that exploits the evolving nature of the networks by considering a simple three-layer graph neural network at each time step and extracting node orientation by using Given’s angle method. Our proposed temporal network-embedding algorithm, TempNodeEmb, is validated by comparing it to seven state-of-the-art benchmark network-embedding models. These models are applied to eight dynamic protein–protein interaction networks and three other real-world networks, including dynamic email networks, online college text message networks, and human real contact datasets. To improve our model, we have considered time encoding and proposed another extension to our model, TempNodeEmb++. The results show that our proposed models outperform the state-of-the-art models in most cases based on two evaluation metrics.

## 1. Introduction

Temporal graphs are amongst the best tools to model real-world evolving complex systems such as human interactions, the Internet, biological interactions, transport networks, scientific networks, and other social and technological networks [1]. Understanding the evolving patterns of such networks has important implications in our daily life, and predicting future links among the nodes in such networks reveals an important aspect of the evolution of temporal networks [2]. To apply mathematical models, networks are represented by adjacency matrices that take into account only the local information of each node and are both high-dimensional and generally sparse in nature. Therefore, they are insufficient for representing global information (e.g., nodes neighbors’ information), which is often an important feature of the network, and consequently cannot be directly used by machine learning (ML) models for predicting graph or node-level changes. Similarly, representing temporal networks using temporal adjacency matrices, as a snapshot of the network at different time steps, involves the same problems and necessitates using alternative methods. This has led to the development of deep neural network-based approaches to learn node/edge level features [3] to be used for graph representation learning. Learning useful representations from networks (or graphs) not only reduces the computational complexity but also provides greater predictive power that facilitates further use of machine learning methods [4]. These representations can be used in various applications such as node classification, link prediction, community detection, and anomaly detection. Additionally, the use of temporal information can also enhance the performance of these applications.

Traditional machine-learning approaches are appropriate for data with Euclidean or grid-like structures, such as image, audio, or text. Graphs, on the other hand, might express non-Euclidean relationships, which typical machine learning models struggle with [5,6]. Additionally, while adjacency matrices change in shape as nodes are added and removed in dynamic graphs and networks, classical techniques have a fixed number of dimensions. Consequently, the traditional ways of solving problems as “supervized” and ”unsupervized” differ when it comes to graphs [7]. With dynamic graphs, where nodes and edges can change over time, these issues become more obvious. These issues are solved by graph-embedding methods that produce low-dimensional and fixed-dimensional vector representations of a graph [8]. The structural features of the graph are extracted and modeled mathematically to achieve this. The resulting fixed-size vector can then be applied to any subsequent tasks, including link prediction, graph classification, and node classification.

Dynamic graph evolution studies have been at the center of network science [9,10,11,12,13,14] and in particular of addressing link prediction challenges [15]. Apart from traditional machine learning (ML) and statistical modeling approaches, deep neural networks are currently also being developed [7]. These models effectively generate a *d*-dimensional feature vector (where *d* is lower than the total number of nodes in a graph) based on the graph structure, which can be used by any ML model for downstream tasks. Consequently, some researchers have come up with matrix factorization approaches [16,17,18], deep neural network auto-encoders [19,20], and convolutional neural networks [21] that often consider random walks [22,23,24].

Until now, few works have used graph embedding, mainly on static graphs or single snapshots of graphs [4,7,25,26,27], for social networks [21,28], for traffic prediction [29,30], for knowledge graphs [31,32], for drug discovery [33,34], and for recommendation systems [35]. However, whereas many real-world applications require time-sensitive forecasts, static graphs do not take into account how graphs behave over time. Examples include when two people interact, consume something (such as online goods or news), human brain neurons form new connections, someone is about to transact, and the road is congested (https://deepmind.com/blog/article/traffic-prediction-with-advanced-graphneural-networks accessed on 1 January 2021) [36]. This demonstrates that the accurate representation of dynamic networks through the development of a powerful dynamic graph-embedding technique and algorithmic predictions will be more useful in solving a number of real-world issues.

The temporal structure of graphs motivates us to look at representation learning techniques for dynamic graphs that can capture the evolutionary characteristics of real-world networks and can be applied to further tasks such as time-varying link prediction and dynamic node classification, among others. In any event, researchers have also solved the problem of inefficient message aggregation over disconnected neighbours from noisy links [37,38]. Because time-varying aggregation propagates noisy data over time, it is a major problem. The model is vulnerable to noisy data due to an over-reliance on graph topologies, which can significantly reduce the accuracy of subsequent prediction tasks, and this poses the need for dynamic graph embedding [39]. In our present work, we are focusing on dynamic graph embedding, which is more complicated than static graphs since new nodes can be added or removed over time [40,41], and sometimes edge or node labels also change over time [42].

The proposed model, TempNodeEmbed, addresses the issue of accurately predicting links in temporal networks. Traditional static-node embedding methods fail to capture the evolution of the graph structure and the interactions between nodes over time. TempNodeEmbed addresses this limitation by incorporating temporal information through a three-step forward operation on a graph neural network and by creating a stable orthogonal alignment between consecutive time steps. Additionally, TempNodeEmbed++ takes into account time encoding and node-level features to improve performance. Through experiments on real-world datasets, TempNodeEmbed and TempNodeEmbed++ have been shown to outperform state-of-the-art methods for link prediction in temporal networks. Thus, the proposed model offers a promising solution for accurately predicting links in dynamic networks. In summary, this research presents a novel deep learning-based model for generating low-dimensional features from large high-dimensional networks considering their temporal information. Our technical contributions are as follows:Instead of a complex static embedding vector-generation method, we developed a simple three-layer graph neural network model without any hyperparameter learning. This simple model considers weighted adjacency, temporal decay effects, and node-level explicit features that are important for generating a node representation in dynamic graphs.Considering a time-varying adjacency matrix, in which entries are $e_{i,j,t}=e^{t-t_{now}}$, where *t* is the time step when the graph was constructed, and $t_{now}$ is the current time. Incorporating this approach enables us to consider: (i) the dynamic nature of the network; (ii) temporal node/edge-level explicit features; and (iii) a weighted edge representation model.Considering angles (using Given’s angle method) between any two consecutive time steps, calculated based on the generated static features.

### Problem Formulation

Graphs are composed of a set of nodes $V=\{v_{1},v_{2}\ldots ,v_{\left|V\right|}\}$ and a set of edges $E_{}=e_{i,j}$ that reflect a connection between each pair of nodes. However, considering dynamic networks, the associated edges $E_{T}=e_{i,j,t}$ contain a time stamp *t*, where i,j,t represents an interaction between node $v_{i}$ and $v_{j}$ at time *t*. So, a dynamic or temporal graph $G_{t}$ can be represented by a three tuple set $G(V,E_{T})$, representing the graph at time *t*, which contains all of the edges that has been formed before time *t*. For training our model, we considered *T* time slices such that t∈[1,T], and used *T* set of temporal graphs $G_{1},G_{2}\ldots ,G_{T}$. Our aim is then to learn a continuous graph-level vector to predict if a link will be formed between two nodes $v_{i}$ and $v_{j}$ at time $T+t^{\prime }$.

The remainder of the paper is organized as follows. We reviewed some related works on node embedding in Section 2. In Section 3, we present our proposed approach for embedding temporal networks: TempNodeEmb. Furthermore, we extended our current model by considering time encoding in Section 3.6. We outline our experimental design, including data sets, evaluation metrics, and benchmark methods, in Section 4, and present the results in Section 5. We close the paper in Section 6 with a discussion and conclusion.

## 2. Related Works

In order to make the application of statistical models more convenient, network embedding, as such a technique, is created for learning hidden representations of network nodes to encode relations in a continuous vector space [23,27]. In other words, network (graph) embedding approaches transform (embed) very large high-dimensional and sparse networks into low-dimensional vectors [43], while integrating the global structure of the network (maintaining the neighbourhood information) into the learning process [16], which has applications in tasks such as node classification, visualization, link prediction, and recommendation [43,44]. Although network-embedding models are best at capturing network structural information, they lack consideration of temporal granularity and fail in temporal-level predictions such as temporal link prediction, and evolving community prediction [45]. The graph embedding in temporal networks for the dynamic or temporal graph problem has received relatively little attention [44,46,47,48,49,50,51,52]. For instance, DYGEM [53] utilizes the learned embeddings from the previous time-step to initialize the embeddings in the following time-step. DYNAERNN [54] applies RNN to smooth node embeddings at various time-steps; (2) recurrent-based techniques catch the time varying dependence utilizing RNN. For instance, GCRN [55] first processes node embeddings on every snapshot by utilizing GCN [56]; then, at that point, it feeds the node embeddings into a RNN to learn their dynamic behaviors. EVOLVEGCN [57] utilizes RNN to calculate the GCN weight boundaries at various time-steps; (3) attention-based techniques utilize the “self-attention” mechanism for both spatial and temporal message aggregation. For instance, DYSAT [58] proposes to utilize the self-attention technique for temporal and spatial data aggregation. TGAT [59] encodes the temporal data into the node embeddings and then, at that point, applies self-attention to the temporal expanded node features.

### 2.1. Random Dot Product Graphs (RDPGs)

The mathematical study of random graphs has its origins in the work of Erdos and R’enyi [60] and E. N. Gilbert [61], who investigated graphs in which edges connecting nodes form independently according to Bernoulli random variables with a fixed probability *p*, in what might be called the simplest probabilistic model of a naturally occurring network (this sort of graph is now referred to as an Erdos–R’enyi graph). Recently, models for random dot product graphs (RDPG) have been brought out in the literature; however, they have not yet been significantly formalized for dynamic graphs. The first examples highlight methods for community detection and clustering [62,63,64]. In recent years, scientists have focused on simulating the brain’s connection networks as random dot product graphs [65,66,67]. To provide discrete representations for each graph and each node, Levin et al. [68] proposed an omnibus embedding by jointly embedding several networks into a single latent space. The multiple random eigen graph (MREG) model, created by Wang et al. [69], has a number of d-dimensional latent properties that are shared by all of the graphs within it. Depending on the network, various weights are applied to the inner product between the latent positions. Another approach, COSIE (common subspace independent edge) [70], has been developed to further expand on this concept. Gallagher et al. [71] use unfolded adjacency spectral embedding (UASE), which was initially proposed for the multilayer random dot product graph (MRDPG) [72], for dynamic graph embedding. The UASE approach is based on the singular value decomposition method of matrix factorization [71]. Gallagher et al. [71] also considered the dynamic latent position model when comparing UASE and other techniques for the task of dynamic network embedding. A link-prediction method for dynamic graphs using RDPG was also presented by Passino et al. [73] for a cybersecurity application.

### 2.2. Learning Node Embedding

Previous approaches have relied on heuristics or hand-engineered techniques such as graph statistics, node-level statistics, and graphlet kernels, which can produce effective results for a single task such as classification. However, in order to solve this issue, automated feature-engineering techniques are needed to develop a fixed-dimensional vector for each node that can be used for all downstream operations. The techniques that have been applied to generate node embeddings are listed below.

### 2.3. Encoder–Decoder Framework for Dynamic Graphs

Hamilton et al. [74] presented an encoder–decoder framework for static graph embedding learning (see, e.g., Figure 1 ($F_{1}$)). The model learns a low-dimensional vector (also known as an encoder) that can be utilized for any downstream task, such as node classification, link prediction, and graph reconstruction. The decoder model is used to perform various downstream tasks; it could be a simple sigmoid function, a traditional machine learning algorithm, or a deep neural network. There are many methods available to learn these low-dimensional vectors [75].

The embedding for dynamic graphs is learned by using these static embeddings at time t<T and extrapolating (>T) or interpolating (<T) at any given time $t^{\prime }$. Most of the problems are related to extrapolation, i.e., $t^{\prime }>T$. The following well-known techniques have been used for learning node embeddings for dynamic graphs.

**Aggregating Temporal Observations:** The simplest method to deal with the dynamic graph embedding is to aggregate all of the adjacency matrices ($A^{t}$) over time *t* into a single adjacency matrix *A* and apply a static graph-embedding technique [75]. This is the first step for dynamic graph embedding [76] but requires aggregation as follows: $A_{aggregate}\left[i\right]\left[j\right]=\sum_{t=1}^{T}A^{t}\left[i\right]\left[j\right]$. Some researchers aggregated using union operations instead of summation [77]. Some researchers considered weight λ∈(0,1) and aggregated it as $A_{aggregate}\left[i\right]\left[j\right]=\sum_{t=1}^{T}\lambda^{T-t}A^{t}\left[i\right]\left[j\right]$ [78,79,80].**Aggregating Static Embedding:** Instead of aggregating whole graphs, some researchers have aggregated and generated embeddings over time. For example, researchers [53,57,58] have made progress in dynamic graph representation learning by learning node representations on each static graph snapshot (at every time step) and then aggregating these representations from the temporal dimension. Let $G^{1},G^{2},\ldots G^{t},\cdots ,G^{T}$ be a snapshot of the graph. In this approach, the embedding is learned every time with respect to graph snapshots $z^{1},z^{2}\ldots z^{t}\ldots ,z^{T}$. Furthermore, $Z^{i}$s are aggregated according to some functions proposed by Yao et al. [81]: $z_{v}=\sum_{t=1}^{T}\exp\left(\lambda \left(T-t\right)\right)z_{v}^{t}$. Zhu et al. [82] aggregated the final embedding as a weighted sum. However, some researchers have applied time series models such as ARIMA, and reinforcement learning approaches instead [83,84,85,86]. Still, these methods are susceptible to noisy data such as missing or spurious links. This error comes from defective message aggregation from unrelated neighbors. Further aggregation over time makes this error more severe when aggregating all of the previous snapshot information over time.**Time as a regularizer:** Another approach can be applied by considering time as a regularizer when regular time interval snapshots exist [81,87,88,89]. A well known regularizer is Euclidian distance based (i.e., $dist\left(z_{v}^{t};z_{v}^{t-1}\right)=∥z_{v}^{t}-z_{v}^{t-1}∥$). However, Singer et al. [47] considered a rotation-based projection approach. Their distance function can be given as $dist\left(z_{v}^{t};z_{v}^{t-1}\right)=∥R^{t}z_{v}^{t}-z_{v}^{t-1}∥$. Furthermore, Milan et al. proposed a regularizer based on the cosine angle between two embedding vectors [90]: $dist\left(z_{v}^{t};z_{v}^{t-1}\right)=∥1-z_{v}^{t}z_{v}^{t-1}∥$.**Decomposition-based encoders:** The decomposition approach is another way of dealing with this problem, in which the temporal snapshot adjacency matrices can be stacked in the form of a tensor, i.e., $B\in R^{\left|\left|V\right|\times \left|V\right|\times T\right|}$. Further, tensor-decomposition approaches can be applied [40]. Yu et al. [91] made use of a time regularizer and predicted future adjacency ${\hat{A}}^{t^{\prime }}$ at any future time $t^{\prime }$ by solving the following optimization problem:
(1)$$\min_{t=T-w}^{T}\sum_{t}e^{-\lambda (T-t)}{∥A^{t}-U{\left(V^{t}\right)}^{\prime }{\left(P^{t}\right)}^{\prime }∥}_{F}^{2}$$
where $P^{t}=\left(1-\beta \right){\left(I-\beta \sqrt{D^{t}}A^{t}\sqrt{D^{t}}\right)}^{-1}$, β∈(0,1) and $U\in R^{\left|V\right|\times d}$.**Random Walk Encoders:** Random walk-based models have been very successful in similarity-based feature representation on static graphs. Mahdavi et al. [44] first generated an evolving random walk for a graph over time, feeding time snapshots at t=1…T to their model by generating random walks for t>T using the (t−1)th snapshot. Bian et al. applied a similar random walk-based technique on a knowledge graph [92]. Furthermore, Sajjad et al. [93] observed that keeping the random walks from previous snapshots shows a different distribution than generating random walks from scratch for every snapshot.**Sequence-Model Encoders:** Another way of solving dynamic network embedding is by applying sequence models using recurrent neural networks (RNN) [56,94,95,96]. Static embeddings are generated for each snapshot and then fed into any of the RNNs to predict the embedding at any time $t^{\prime }$ in the future. As RNNs can work asynchronously or synchronously, these approaches are well-utilized.**Autoencoder-based Encoders:** Kamra et al. [53] used an auto-encoder (AE)-based embedding, learning $AE^{t}$ (i.e., auto-encoder at time t) for $G^{t}$ (i.e., graph at time t) to generate $z_{v_{i}}^{1}$ for node $v_{i}$. If $z_{v_{i}}^{1}$ and $z_{v_{j}}^{1}$ are linked together, they are constrained to be close in the embedding space. To achieve node addition, they used a heuristic-based method considering previous snapshots to enable the learning of an auto-encoder for the current snapshots. Furthermore, to have better embedding, Goyal et al. [54] considered all previous snapshots for learning the embedding at current snapshots. Additionally, Rahman et al. [97] followed an AE-based approach by considering node pairs instead of single nodes. This approach helped them with learning representation for edge addition and deletion problems.**Diachronic Encoders:** Most of the previous methods map either nodes or edges to hidden representations, but diachronic encoders map every pair of nodes and time-stamp to a hidden representation. This makes diachronic encoders a better choice for dynamic graph embedding. Xu et al. and Dasgupta et al. [98,99] proposed diachronic encoder models that consider time as a parameter of embedding functions, while Goel et al. [100] proposed a diachronic encoder for knowledge graph embedding where $z_{v}^{t}\in R^{d}$ is a function of time *t*.

## 3. Materials and Methods: Our Proposed TempNodeEmbed Model

In this section, we present and discuss our proposed solution for graph representation learning to assist link prediction in dynamic networks. To develop a temporal graph representation, we first generate a *d*-dimensional continuous feature vector for every node, at each time, and then use gated recurrent unit (GRU) [101] for semi-supervized prediction tasks. The detailed processes of our proposed framework (see Figure 2 also pseudo code Algorithm 1) are discussed below:
**Algorithm 1** TempNodeEmbed ($G_{1},G_{2}\cdots ,G_{T}$, where $G_{T}=G\left(V,E_{T}\right)$,$V=\{v_{1},v_{2}\ldots ,v_{\left|V\right|}\}$),**Require:** Input: $G_{1},G_{2}\ldots ,G_{T}$
Step 1. Generate $X_{1},X_{2},\ldots ,X_{T}$, where $X_{t}\in R^{\left(\left|V\right|\times d\right)}$ are latent feature matrices. Each node *v* has a historical embedding of size *d*. These matrices take into account explicit temporal node-level features as well.Step 2. For TempNodeEmbed++, use the softmax nonlinearity in Step 1 and concatenate time encoding.Step 3. Find the orthogonal basis matrices between two consecutive time steps by applying the orthogonal procrustes theorem.Step 4. Use these orthogonal basis matrices to generate the next time step embedding using a learnable function $L_{T}$. The function is learned by minimizing a task-oriented cost function.Step 5. To learn the embedding pattern, we use a recurrent neural network with a gating mechanism (gated recurrent unit), which uses historical *d*-dimensional node embeddings for temporal pattern learning and can be used to generate node embeddings at any time t>T.


### 3.1. Graph Neural Network Operation

At every time step *t* from the training set, we generate a *d*-dimensional feature vector for every node ($d\ll \left|V\right|$, where $\left|V\right|$ is the number of nodes in *G*), by applying the following operations. We assume that in the temporal graph domain, the embeddings of two graphs $G_{t_{i}}$ and $G_{t_{j}}$ are carried out individually; hence, it is not guaranteed that the node embeddings will remain the same even if the graphs are similar over the time points $t_{i}$ and $t_{j}$. Therefore, we generate static embeddings independently for each time step. For a given time *t*, the temporal adjacency matrix is represented as $A{}_{i,j}^{t}$ (which can be weighted), and the temporal influence matrix, ${{\hat{A}}^{t}}_{e}$, can be formulated as
(2)$${{\hat{A}}^{t}}_{e}=e^{t-\left(t_{now}+\epsilon \right)}\times \left(A{}_{i,j}^{t}+I\right)$$
where *I* is an identity matrix, it has only diagonal elements that are 1 (representing only self-loops: node *i* links to itself), and ϵ is an arbitrarily low value (0.00001) to map binary values to a number less than 1.

Suppose we have a matrix $A_{t}$ at time *t* with size $\left|V\right|\times \left|V\right|$ (built from a graph structure). We introduce a self-loop by adding an identity matrix I;${\hat{A}}_{t}=A_{t}+I$.The temporal edge matrix will be ${{\hat{A}}^{t}}_{e}=e^{\left(t-\left(tnow+\epsilon \right)\right)}\cdot {\hat{A}}_{t}$

We assume that a node’s edge influence decreases exponentially while considering its temporal influence.

### 3.2. Generating Static Embedding

In order to develop fundamental conclusions on prediction for dynamic networks, we focus on a particular subclass of random graph models known as latent position random graphs [102]. By providing each node by a typically hidden vector in some low-dimensional Euclidean space $R^{d}$, edges between nodes subsequently develop independently in such graphs. Network inference is transformed into the recovery of lower-dimensional structure in latent position random graphs, which have the appealing property of modelling network connections as functions of inherent properties of the nodes themselves. These features are recorded in the latent positions. More exactly, each network is associated with a matrix $X_{t}$ whose rows are the latent vectors of the nodes if we have a collection of time-indexed latent position graphs $G_{t}$ on a shared aligned node-set. The probabilistic evolution of the network time series is entirely governed by the evolution of the rows of $X_{t}$ because the edge formation probabilities are a function of pairings of rows of $X_{t}$. The rows of $X_{t}$ are thus the obvious subject of investigation for drawing conclusions about a time series of latent position graphs. Anomalies or change points in the time series of networks, in particular, correlate to modifications in the $X_{t}$ process. For instance, a change in a particular network entity is connected to a change in its estimated latent position.

At every time step, we generate a static *d*-dimensional embedding $\in X_{t}$ for every node *v*, using a three-layer of graph neural network as follows. We generate a static embedding matrix $X_{t}$ at every time step *t*, in which the simplest GNN forward propagation model (presented below) is used: (3)$$f{\left(R^{l},A\right)}_{t}={{\hat{A}}^{t}}_{e}{R^{l}}_{t}{W^{l}}_{t}$$
where $R^{l}$ is a hidden representation, $W^{l}$ is a random weight matrix at layer *l*, and $R^{0}=I_{h}$ ($I_{h}$ is a one-hot vector in case when there are no explicit features available for each node. Otherwise, $R^{0}$ is initialized with node-level explicit features, say $F^{0}$). It is noteworthy that we neither apply the degree matrix normalization technique [21] nor any non-linear activation function in this model. These steps are used to generate a static node embedding ($X_{t}^{n\times d}$) at each time step *t*.

Once we have generated a static embedding for each node at each time step, we have a matrix similar to a latent position matrix $X_{t}\in R^{\left(n\times d\right)}$. So, we have $X_{0},X_{1},\cdots ,X_{t},\cdots ,X_{T}$ latent matrices at each time step. Furthermore, these static embeddings are fed into recurrent neural networks for task-dependent embedding learning.

### 3.3. Calculating Node Alignment

Finding node alignments across time is one of the key tasks in embedding temporal networks. In this work, we calculate how the specific attributes of nodes change rather than computing the angles between two nodes. We analyze the angle between features at two separate time steps as defined by angles between two scalars when two features, at times *t* and t+1, lie in the same Euclidean space [103].

Using the two static feature matrices Xt and Xt+1 (Equation  (Equation 3)) of a graph at times *t* and t+1, respectively.

Our goal is to reduce the difference between two time steps, $t_{i}$ and $t_{j}$, which come from several embedding training sessions. We perform an orthogonal transformation between the node embeddings at time $t_{i}$ and the node embeddings at time $t_{j}$ under the assumption that the majority of nodes have not changed significantly between $t_{i}$ and $t_{j}$. We employ the orthogonal procrustes method, which approximates two matrices using least-squares methods. Let $X_{t}\in R^{n\times d}$, as applied to our problem, be the matrix of node embeddings at time step *t*. Iteratively, we align the matrices corresponding to the subsequent time steps, first aligning $X_{2}$ to $X_{1}$ and then $X_{3}$ to $X_{2}$, and so on. Finding the orthogonal matrix $Q_{t}$ between Xt and Xt+1 is necessary for alignment. The following regression problem is optimized to produce the approximation:(4)$$Q_{t+1}=argmin_{Qs.t.Q^{T}Q=I}∥QX_{t+1}-X_{t}∥$$
where $Q_{t}\in R^{d\times d}$ is the optimal orthogonal alignment between the two consecutive time steps.

Further, we have found an optimized solution as follows; we calculate the angle between its individual features using Algorithm 2. In order to know how each feature aligns over time, we create matrices $\Theta_{\cos\alpha }$ and Θcosβ. Furthermore, we apply dot operations, i.e., matrix $C_{t}=\Theta^{T}\cos\beta \cdot \Theta_{\cos\alpha }$. To find a stable matrix between any two consecutive snapshots, we decompose the $C_{t}$ matrix as $C_{t}=Q_{t}\;*\;R_{t}$ (using the QR decomposition method because $C_{t}$ is a square matrix).
**Algorithm 2** Calculating the angles($x_{v}\left(t,i\right)$),$x_{v}\left(t+1,i\right)$**Require:** Input: $x_{v}\left(t,i\right)$,$x_{v}\left(t+1,i\right)$
**if**$x_{v}\left(t+1\right)=0$**then**   cosα=1;cosβ=0**else**   **if** $|x_{v}\left(t+1,i\right)|>|x_{v}\left(t,i\right)|$ **then**     $tmp=-x_{v}\left(t,i\right)/x_{v}\left(t+1,i\right)$     $cos\beta =1/\sqrt{1+tmp^{2}}$     cosα=tmp.cosβ   **else**     $tmp=-x_{v}\left(t+1,i\right)/x_{v}\left(t,i\right)$     $cos\alpha =1/\sqrt{1+tmp^{2}}$     cosβ=tmp.cosα   **end if****end if**Output: cosα,cosβ


### 3.4. Loss Function

Our aim is to learn feature vector at time step *T* using function $l_{T}\left(v\right)$. For temporal link prediction tasks, we learn the parameters using cross-entropy loss, as follows:(5)$$Cost\left(p,\hat{p}\right)=-p\log\left(\hat{p}\right)-\left(1-p\right)\log\left(1-\hat{p}\right)$$
where *p* is the actual label and $\hat{p}$ is the predicted label. In our link-prediction problem, we have considered function *C* as the concatenation function between features of node $v_{1}$ and node $v_{2}$. As link-prediction tasks happen between two nodes, we used the concatenation function. Furthermore, given graph snapshots $G_{1},G_{2},\ldots ,G_{T}$, we learn the function $L_{T}$ by minimizing the cost $Cost(p,\hat{p})$ for link prediction, as follows:(6)$$l_{T}(v)=L_{T}\left(v,G_{1},G_{2},\ldots ,G_{T}\right)$$The function $l_{T}\left(v\right)$ is used to learn the node embeddings in a temporal graph by combining the embeddings of the nodes at each time step into a single, final embedding. This allows the node embeddings to capture the temporal evolution of the graph structure and the interactions between nodes over time. Finally, we learn the final orientation using a recursive function, as described by Singer et al. [47] as follows:(7)$$l_{t+1}\left(v\right)=\sigma \left(Al_{t}\left(v\right)+BQ_{t}X_{t}v\right)$$
where $l_{0}\left(v\right)=\vec{0}$, $A,B,Q_{t}$ are matrices that are learned during training and σ is the activation function. In our case, we use the tanh function.

### 3.5. Learning for Link Prediction

After obtaining *d*-dimensional stable aligned vectors for each node at each time, we use gated recurrent units (GRUs) [101] for training the network by formulating our link-prediction problem as a binary classification problem. Furthermore, the generated node features of any two nodes are concatenated so that the neural network can learn the probability scores of having a link between any two nodes.

### 3.6. TempNodeEmbed++: Further Extension of Our Proposed Model

Furthermore, we have concatenated time encoding [59] while generating static embeddings. Additionally, we have applied a soft-max activation function (imposing non-linearity) while generating static embeddings as follows:(8)$$f\left(R^{l},A\right)={\hat{A}}_{e}^{t}softmax\left(R^{l}W^{l}\right)$$The time encoding is concatenated to include temporal effects more effectively.

## 4. Experimental Design

In order to evaluate and compare the performance of different methodologies, we used several temporal network datasets. The data were split into two parts based on a pivot time, with 80 percent of the edges used for training and the remaining 20 percent for testing. The basic properties of the datasets are shown in Table 1. For the training set, all edges that were created at or before the pivot time were considered as positive examples. All edges that were created after the pivot time but before the test time were considered as positive test examples. To create negative examples, a similar number of edges were randomly sampled. We randomly sampled the same number of edges from all node pairs that were not connected at pivot time for the training set’s negative examples as we did for the positive ones. For the test set’s negative examples, we randomly selected the same number of edges from all node pairs that were not connected by any edges at all. To evaluate our model, the number of nodes in the hidden layers is randomly selected as the number of nodes in the graph divided by 2. The number of neurons in the final layer is the number of dimensions we want to keep for each node, which we set to 128. For other models that require manual parameter tuning, such as node2vec and DeepWalk, we kept the default parameters used in the library. We used the open-source Cogdl Python library (https://github.com/THUDM/cogdl accessed on 31 January 2021) to implement our model and the baselines.

### 4.1. Datasets

The effectiveness of our approach is assessed using the real-world datasets listed below, which are excellent examples of dynamic graphs:**Protein–protein interaction (PPI) network:** This includes proteins as nodes and an edge between any pairs of proteins that are biologically interacted with. The interaction-discovery dates are considered the edge’s timestamp. A yearly granularity between 1970 and 2015 is used as time steps in this dataset [47].**Dynamic protein–protein interaction (DPPIN) network:** We use 7 dynamic protein–protein interaction networks of yeast cells at different scales, including Yu, Ho, Tarassov, Lambert, Krogan-MALDI, Krogan-LCMS, and Babu, published by Fu et al. [104]. These datasets were created by the following these steps: (1) identifying the active gene-coding proteins at a given timestamp; (2) identifying the co-expressed protein pairs at that timestamp; and (3) preserving only the active and co-expressed proteins for dynamic protein interactions at that timestamp [104].**Dynamic email network (EU-Email):** Significant European research institutions’ email data were used to create the network, as mentioned in [105]. The identities of the sender and recipient are anonymized. The network is composed of email interactions between individuals at the institutions over a period of time. The interactions are represented as edges between individuals, with the edge representing an email exchange between the two individuals. The edges are directed, with the sender as the source node and the recipient as the target node. The data also include timestamps for each email exchange, allowing for the analysis of the dynamic nature of the interactions over time.**MIT human contact (MITC) network:** (from [106]) This undirected network contains human-contact data among students of the Massachusetts Institute of Technology (MIT), collected by the Reality Mining experiment performed in 2004 as part of the Reality Commons project [107]. A node represents a person, and an edge indicates that the corresponding nodes had physical contact. The data were collected over a period of 9 months using mobile phones. For time steps in this dataset, a daily granularity is used.**College text message (COLLMsg) network:** Data were collected from a social networking app, similar to Facebook, used at the University of California, Irvine. The nodes in the network represent individuals, and a directed edge represents a message sent from one user to another. The time steps in this dataset have daily granularity, with data collected between 15 April 2004 and 26 October 2004.

### 4.2. Evaluation Metrics

Two common machine learning assessment metrics, AUPR and AUROC, are employed and are defined as follows:

**Precision:** The percentage of true positives compared to all positives is how precision is measured. For $T_{P}$ items that were correctly predicted as positive and $F_{P}$ items that were incorrectly predicted as positive (i.e., false positives), the formula for precision is:(9)$$\mathrm{Precision}=\frac{T_{P}}{T_{P}+F_{P}}\;.$$The “recall” metric, which penalizes the score with false negatives, is used to measure the misclassification of actual positives. Recall is defined as, if $F_{N}$ is the number of false negatives,
(10)$$\mathrm{Recall}=\frac{T_{P}}{T_{P}+F_{N}}.$$

The false positive rate (FPR) is calculated as
(11)$$FPR=\frac{F_{P}}{\left(T_{N}+F_{P}\right)},$$
where $F_{P}$ is the number of false positives and $T_{N}$ is the number of true negatives. **AUROC**: The true positive rate (TPR) and the false positive rate are plotted against one another, and the area under that line is known as the area under the receiver operating characteristicss (AUROC) value (FPR). The trade-off between TP and FP prediction rates is represented by it. The chance of detection, sensitivity, or recall are further terms for the TPR. AUROC is a crucial metric because it assesses the classifier’s separability.

**AUPR**: The precision and recall accuracy are simultaneously estimated using the area under the precision and recall (AUPR) curve. In other words, changing threshold levels affects how the precision-recall pair points are calculated. This indicator shows how well the models can handle skewed distributions and predict efficiency when there are imbalanced classes.

### 4.3. Optimization Algorithm

We employ the Adam optimizer [108], which computes an exponentially weighted average of previous gradients and eliminates biases, for parameter learning.

#### Baseline Methods

In order to evaluate its performance, we compared our proposed model to several state-of-the-art temporal embedding and static-node embedding methods. While the dynamic model utilizes all previous snapshots taken before or at time *t*, the static techniques use only the network snapshot taken at time *t* to make predictions for t+1.

**tNodeEmbed [47]**: This method is the state-of-the art for node embedding for dynamic graphs. It learns embedding by first generating static embedding and then finding node alignments. Furthermore, it is fed to a recurrent neural network for task-oriented predictions.**Dyngraph2vecAE [54]**: This method is also state-of-the-art for node embedding for dynamic graphs. This method learns node embedding using an auto encoder and a recurrent neural network.**Prone [109]**: This method first initializes the embedding using sparse matrix factorization and spectral analysis for local and global structural information.**DeepWalk [23]**: This model learns a node’s low dimensional embedding based on random walks. It has two hyper-parameters: walk length *l*, and window size *w*.**Node2vec [24]**: It is a similar model for graphs that works on similar principal of Word2vec model [110], as a framework for word embedding in natural language processing. Based on Word2vec’s related skip-gram notion. It generates low-dimensional embedding and operates on neighbourhood nodes. Node2vec can be generalized depending on the situation, for example, if one wants to include similarities based on location or on a node’s function in a network.**LINE [43]**: By taking into account first-order and second-order node similarity, this model creates node low dimesional embedding. The performance of this model is also enhanced for large-scale networks by the use of sampling based on edge weights. It is a DeepWalk special case when the size of the nodes context is kept at 1.**Hope [17]**: The Katz index and PageRank are the foundations of the high-order proximity preserved embedding technique. Low-rank approximations are made using the singular value decomposition technique.

Basic dataset attributes, such as the number of nodes, links, or weighted or binary representations, are provided in Table 1. The code for our suggested model is now accessible online at GitHub for reproducibility (https://github.com/khushnood/TempNodeEmbed_upload accessed on 25 January 2023).

## 5. Experimental Results

To evaluate the performance of our proposed dynamic link prediction model (“TempNodeEmbed”), we compared it to seven baseline models on several real-world datasets. The results are reported in Table 2, Table 3, Table 4 and Table 5. Our model exhibited the most reliable performance, obtaining the best outcome across all eleven datasets. The performance outcomes and the deviation from the baselines vary significantly among the datasets.

### 5.1. Performance Evaluation on Link Prediction Task

Our proposed model (TempNodeEmbed) outperforms all of the baseline models, as demonstrated by the results in Table 2 and Table 3. It is noteworthy that we have presented our model in its most basic version, requiring no hyperparameter tuning for the creation of static embeddings. It is superior to tNodeEmbed and other models that do not take into account node-level features as it also considers the weighted adjacency matrix and explicit node-level features. Additionally, our proposed TempNodeEmbed++ (see Section 3.6) has been shown to be effective, as demonstrated by the results in Table 4 and Table 5. With a significant margin, this model outperforms all of the baseline models. We have found that incorporating a time-encoding strategy improves the performance of our model on additional datasets.

### 5.2. Nodel Alignment Analysis

In this section, we demonstrate the optimization capability of our framework when using Algorithm 2. We propose a new method for the Procrustes theorem and have found, through empirical analysis, that our schema improves the algorithmic performance. To evaluate the performance, we compared our proposed Procrustes method to the one used in [47]. We conducted experiments 10 times and compared the results.

Figure 3 compares the area under the receiver operating characteristics (AUROC) scores for the two Procrustes methods, labeled “Node Alignment (Old)” (reported in [47]) and “Node Alignment (Proposed)” (see Section 3.3). The x-axis lists different datasets, including PPI, Yu, Tarassov, Lambert, MALDI, LCMS, Ho, and Babu. The y-axis shows the ROC scores, with a range of 0 to 0.9. The “Node Alignment (Proposed)” model generally has higher ROC scores than the “Node Alignment (Old)” model across all datasets. The similar pattern is also seen for the area under the precision-recall (AUPR) score. This result proves that our proposed node alignment method improves the overall performance of the framework.

### 5.3. Effect of Embedding Vector Size

We encode a node’s information into a fixed-size vector (*d*). The model’s capacity for prediction is impacted by this fixed size. For instance, if the vector size is kept very small, certain information is left out. To effectively embed the node information, a lower bound (i.e., the smallest vector size) should exist. An algorithm would need a small vector size to effectively encode node/edge or graphs into a continuous vector. We ran an experiment on a number of datasets with various embedding vector sizes to gauge this capacity. In Figure 4, we presented the outcomes of two analyses along with our standard performance measures (AUROC and AUPR) and their standard deviations (SD). Initially, when the vector size is 2, there is a lot of fluctuation in the results, but as the vector size is increased, the SD drops and stabilizes. The accuracy results show a trend that is comparable. This shows that in order for our model to perform better across all datasets, it is necessary to determine the ideal vector size, which suggests that below a particular threshold vector size our model’s performance will be affected negatively.

### 5.4. Effect of GNN Layers

We empirically analyzed the effect of GNN layers on the performance of our model. To do this, we randomly selected four datasets and varied the number of GNN layers from 2 to 8. We observed that after 3–4 layers, the results did not improve, as seen in Figure 5. This is known as the over-smoothing problem in GNN. When the network becomes deeper, every node has similar features due to the message passing at each layer, resulting in each node having the same feature representation. This is why GNNs perform better with shallow networks. Based on these results, we only considered three layers in our work to keep the model simple, although finding the best architecture could potentially result in improved performance. Finding the best GNN architecture is an active research area (see references [111,112]), and many researchers agree that shallow networks perform better.

## 6. Conclusions

In this study, we presented a highly efficient and simple model for generating node embeddings in temporal or dynamic graphs. To achieve this goal, we created a temporal effect matrix and a static embedding of nodes at each time step using a feed-forward three-step operation on a graph neural network. The most significant distinction is that we produced a static embedding that is unsupervized and does not require any non-linear activation functions. Even just a three-step forward propagation operation improves performance. Additionally, our model takes into account changing node properties when creating static embeddings. In our proposed model, time encoding has also been taken into account. We called it TempNodeEmbed++, which proved to be better than the original TempNodeEmbed and other baseline models. We performed experiments on three real-world datasets, namely, the EU-Email, COLLMsg, and MITC datasets. We found that TempNodeEmbed++ outperforms all of the baselines on AUC and AUPR metrics. On the MITC dataset, dyngraph2vecAE was unable to produce results. Additionally, on the MITC dataset, the TempNodeEmbed model outperforms TempNodeEmbed++, which suggests that not all datasets require nonlinear activation. Sometimes, a simpler model can produce better results.

One limitation of this study is that it only considered growing networks and did not perform any experiments on datasets involving node removal. This should be addressed in future work. Additionally, while our model outperforms state-of-the-art methods, further efforts can be made to improve its efficiency as the process of learning static feature vectors and alignment at each time-step requires more computational resources than models for static graphs. It should also be noted that for the PPI dataset used in this study, node-level explicit features were not available, so we initialized features as one-hot vectors. Despite this, our model still performed better than the tNodeEmbed and dyngraph2vecAE models. All other datasets used in this study have node-level features.

## Figures and Tables

**Figure 1 entropy-25-00257-f001:**
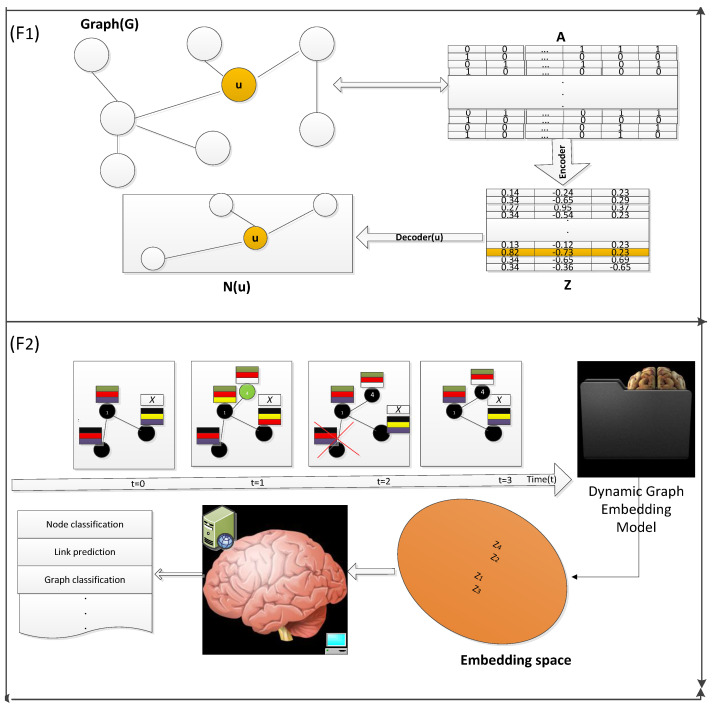
(F1) How graph embedding is generated and re-used for the reconstruction of the graph. It takes the graph, G, as input in the form of an adjacency matrix, A. Furthermore, a function, namely the encoder, generates a corresponding embedding matrix, Z. See how node u has changed its representation vector to a continuous value representation vector (of the matrix Z). Using Z, a matrix decoder can perform any required task, such as link prediction and neighborhood reconstruction. For example, we have described the neighborhood reconstruction for the highlighted (yellow) node, u. (F2) How the dynamic graph-embedding method works. In F2, we can see nodes changing their features differently at different times, we have shown it by varying different color vectors. The direction of the arrow shows time evolution.

**Figure 2 entropy-25-00257-f002:**
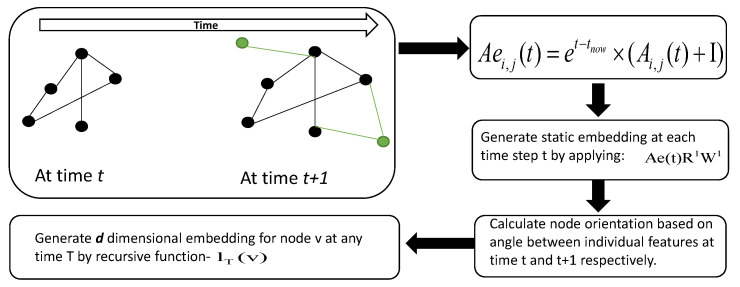
This is the proposed model framework for generating *d*-dimensional node embeddings for temporal graphs. The green nodes represent newly added nodes in the graph.

**Figure 3 entropy-25-00257-f003:**
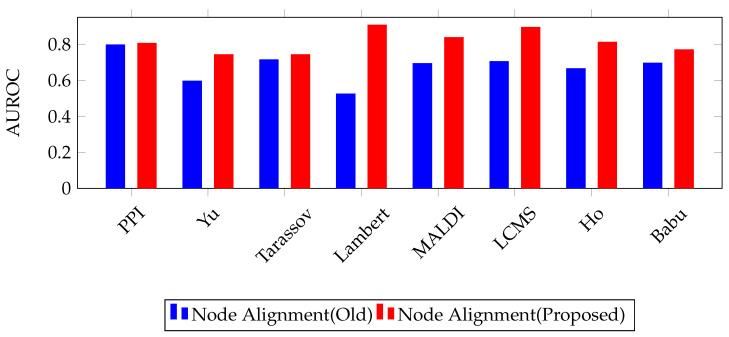
AUROC scores when using node alignment in proposed Vs tNodeembed model on all datasets.

**Figure 4 entropy-25-00257-f004:**
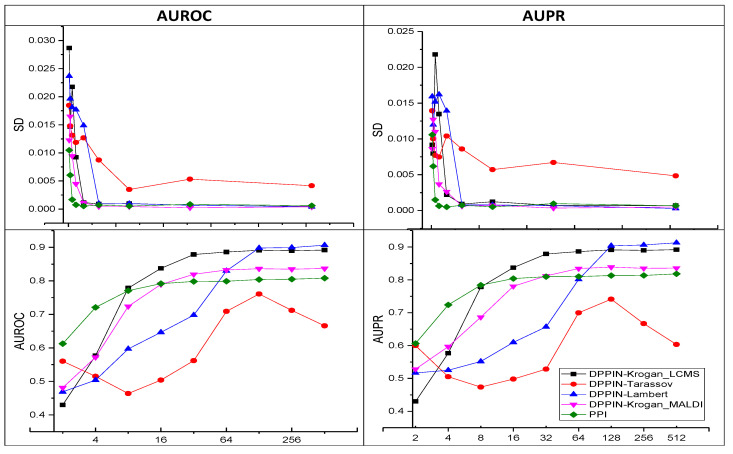
Impact of various embedding vector sizes on the performance of our model. The standard deviation (SD) is displayed in the first row, and the x-axis of the second row shows the AUROC and AUPR vector sizes.

**Figure 5 entropy-25-00257-f005:**
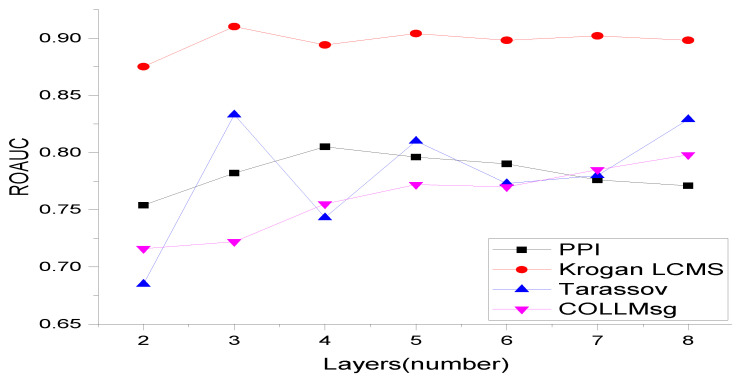
In this figure, we have varied GNN layers described in Equation (Equation 3).

**Table 1 entropy-25-00257-t001:** Basis properties of the data-sets used in our study.

Dataset	Nodes	Edges	Weighted	Node Level Feature
PPI	16,458	144,033	No	No
Lambert	697	6654	Yes	Yes
Tarassov	1053	4826	Yes	Yes
Yu	1163	3602	Yes	Yes
Ho	1548	42,220	Yes	Yes
Krogan_MALDI	2099	78,297	Yes	Yes
Krogan_LCMS	2211	85,133	Yes	Yes
Babu	5003	111,466	Yes	Yes
EU-EMAIL	986	16,064	No	No
MITC	96	2539	No	No
COLLMsg	1899	59,835	No	No

**Table 2 entropy-25-00257-t002:** The table shows the comparison of temporal link prediction on the AUROC metric. Results are shown in the form of “mean ± standard deviation”.

Datasets	TempNodeEmbed	TempNodeEmbed (Static)	dyngraph2vecAE	tNodeEmbed	Deepwalk	Hope	Line	Node2vec	Prone
PPI	**0.805 ± 0.0091**	0.677 ± 0.035	0.722 ± 0.043	0.753 ± 0.006	0.702 ± 0.001	0.766 ± 0.003	0.712 ± 0.002	0.719 ± 0.003	0.762 ± 0.002
Yu	**0.743 ± 0.023**	0.629 ± 0.029	0.615 ± 0.663	0.621 ± 0.001	0.548 ± 0.034	0.546 ± 0.021	0.509 ± 0.023	0.582 ± 0.018	0.540 ± 0.030
Tarassov	**0.743 ± 0.042**	0.626 ± 0.059	0.694 ± 0.017	0.566 ± 0.003	0.594 ± 0.040	0.510 ± 0.045	0.730 ± 0.013	0.673 ± 0.034	0.540 ± 0.017
Lambert	**0.907 ± 0.013**	0.600 ± 0.042	0.775 ± 0.026	0.632 ± 0.018	0.645 ± 0.0025	0.693 ± 0.034	0.727 ± 0.019	0.666 ± 0.013	0.635 ± 0.036
Krogan_MALDI	**0.837 ± 0.003**	0.644 ± 0.036	0.815 ± 0.008	0.687 ± 0.007	0.754 ± 0.004	0.769 ±0.004	0.835 ± 0.002	0.760 ± 0.004	0.745 ± 0.002
Krogan_LCMS	**0.894 ± 0.003**	0.634 ± 0.006	0.881 ± 0.006	0.793 ± 0.012	0.783 ± 0.010	0.854 ± 0.0042	0.841 ± 0.006	0.771 ± 0.006	0.803 ± 0.007
Ho	**0.812 ± 0.011**	0.626 ± 0.049	0.551 ± 0.028	0.633 ± 0.011	0.631 ± 0.008	0.636 ± 0.009	0.766 ± 0.004	0.652 ± 0.009	0.586 ± 0.005
Babu	**0.769 ± 0.003**	0.637 ± 0.031	0.729 ± 0.012	0.662 ± 0.010	0.713 ± 0.004	0.711 ± 0.002	0.755 ± 0.002	0.689 ± 0.004	0.703 ± 0.002

**Table 3 entropy-25-00257-t003:** The table shows the comparison of temporal link prediction on the AUPR metric. Results are shown in the form of “mean ± standard deviation”.

Datasets	TempNodeEmbed	TempNodeEmbed (Static)	dyngraph2vecAE	tNodeEmbed	Deepwalk	Hope	Line	Node2vec	Prone
PPI	**0.814 ± 0.008**	0.677 ± 0.02	0.732 ± 0.026	0.758 ± 0.007	0.692 ± 0.002	0.782 ± 0.002	0.713 ± 0.001	0.714 ± 0.003	0.765 ± 0.002
Yu	**0.755 ± 0.021**	0.632 ± 0.034	0.663 ± 0.03	0.614 ± 0.015	0.559 ± 0.041	0.550 ± 0.0021	0.532 ± 0.024	0.576 ± 0.02	0.533 ± 0.023
Tarassov	0.719 ± 0.066	0.605 ± 0.059	0.70 ± 0.013	0.563 ± 0.039	0.559 ± 0.037	0.508 ± 0.037	**0.761 ± 0.012**	0.629 ± 0.041	0.534 ± 0.020
Lambert	**0.914 ± 0.008**	0.567 ± 0.023	0.771 ± 0.024	0.625 ± 0.020	0.625 ± 0.0030	0.686 ± 0.039	0.730 ± 0.009	0.618 ± 0.016	0.617 ± 0.041
Krogan_MALDI	**0.838 ± 0.005**	0.592 ± 0.027	0.810 ± 0.011	0.649 ± 0.010	0.755 ± 0.006	0.797 ± 0.004	0.837 ± 0.002	0.772 ± 0.005	0.756 ± 0.003
Krogan_LCMS	0.895 ± 0.003	0.586 ± 0.064	**0.901 ± 0.003**	0.788 ± 0.016	0.769 ± 0.017	0.881 ± 0.002	0.851 ± 0.005	0.760 ± 0.007	0.825 ± 0.007
Ho	**0.811 ± 0.013**	0.581 ± 0.032	0.568 ± 0.019	0.589 ± 0.010	0.607 ± 0.006	0.633 ± 0.009	0.733 ± 0.005	0.618 ± 0.011	0.604 ± 0.010
Babu	**0.792 ± 0.004**	0.590 ± 0.027	0.775 ± 0.011	0.678 ± 0.013	0.745 ± 0.006	0.754 ± 0.002	0.797 ± 0.001	0.715 ± 0.006	0.742 ± 0.003

**Table 4 entropy-25-00257-t004:** Results for TempNodeEmbed++ on AUC metric.

Models	TempNodeEmbed++	TempNodeEmbed	dyngraph2vecAE	tNodeEmbed	Deepwalk	Hope	Line	Node2vec	Prone
EU-Email	**0.821 ± 0.007**	0.699±0.053	0.770±0.009	0.596±0.008	0.619±0.011	0.666±0.011	0.672±0.013	0.595±0.012	0.667±0.011
COLLMsg	0.802±0.007	0.756±0.013	0.753±0.011	0.594±0.018	0.519±0.013	0.634±0.010	0.556±0.013	0.536±0.014	0.615±0.009
MITC	0.755±0.048	0.788±0.050	NA	0.604±0.052	0.708±0.039	0.691±0.045	0.619±0.027	0.615±0.036	0.695±0.058

**Table 5 entropy-25-00257-t005:** Results for TempNodeEmbed++ on AUPR metric.

Models	TempNodeEmbed++	TempNodeEmbed	dyngraph2vecAE	tNodeEmbed	Deepwalk	Hope	Line	Node2vec	Prone
EU-Email	**0.817 ± 0.008**	0.630±0.064	0.768±0.011	0.572±0.015	0.579±0.011	0.651±0.013	0.651±0.017	0.559±0.011	0.653±0.008
COLLMsg	0.790±0.007	0.755±0.015	0.746±0.008	0.588±0.015	0.506±0.012	0.611±0.010	0.529±0.011	0.531±0.014	0.603±0.007
MITC	0.741±0.041	0.754±0.069	NA	0.557±0.052	0.682±0.040	0.656±0.050	0.604±0.028	0.596±0.037	0.676±0.065

## Data Availability

In this section, The code for this project is available on our GitHub repository and is fully reproducible using the provided template data. We encourage others to use and build upon our work, and we make every effort to ensure that our code is easy to understand. Please find the link to the GitHub repository: https://github.com/khushnood/TempNodeEmbed_upload.

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
