# Peer review of "A Novel Temporal Network-Embedding Algorithm for Link Prediction in Dynamic Networks"

_entropy, 2023, doi:10.3390/e25020257_

Round 1

Author Response

Original Manuscript ID: entropy-2129241  

Original Article Title: “A Novel Temporal Network Embedding algorithm for Link Prediction in Dynamic Networks”

To: Entropy Editor

Re: Response to reviewers

Dear reviewers;

We would like to express our gratitude to all of the reviewers for providing us with the opportunity to edit our submission and for their insightful criticism and recommendations. Their feedback has allowed us to carefully reconsider a number of crucial aspects of this work. As a result, we have addressed their suggestions and made the best revisions and improvements possible to the article. In the paragraphs that follow, we will elaborate on our responses to the reviewers' criticisms and outline the adjustments that were made in response.

Dear Editor,

Thank you for allowing a resubmission of our manuscript, with an opportunity to address the reviewers’ comments.

We are including two files: (a) a detailed answer to the reviewers' comments (response to reviewers), (b) an updated manuscript.

Best regards, Dr. Abbas and coauthors. 

Author Response

Original Manuscript ID: entropy-2129241  

Original Article Title: “A Novel Temporal Network Embedding algorithm for Link Prediction in Dynamic Networks”

To: Entropy Editor

Re: Response to reviewers

Dear reviewers;

We would like to express our gratitude to all of the reviewers for providing us with the opportunity to edit our submission and for their insightful criticism and recommendations. Their feedback has allowed us to carefully reconsider a number of crucial aspects of this work. As a result, we have addressed their suggestions and made the best revisions and improvements possible to the article. In the paragraphs that follow, we will elaborate on our responses to the reviewers' criticisms and outline the adjustments that were made in response.

Best regards,

Dr. Abbas and coauthors. 

Round 2

Reviewer 1 Report

As all my concerns have been properly addressed by the authors, I recommend the manuscript be published in Entropy.

Reviewer 2 Report

My questions have been well answered, and I have no other questions. I recommend this paper for publication, thank you!